# Hydrogen-bonded organic frameworks in solution enables continuous and high-crystalline membranes

Qi Yin[1,4], Kuan Pang[1,2,4], Ya-Nan Feng[1], Lili Han[1], Ali Morsali[3], Xi-Ya Li[1] & Tian-Fu Liu [1,2] ✉

Hydrogen-Bonded organic frameworks (HOFs) are a type of emerging porous materials. At present, little research has been conducted on their solution state. This work demonstrates that HOFs fragment into small particles while maintaining their original assemblies upon dispersing in solvents, as confirmed by Cryo-electron microscopy coupled with 3D electron diffraction technology. 1D and 2D-Nuclear Magnetic Resonance (NMR) and zeta potential analyses indicate the HOF-based colloid solution and the isolated molecular solution have significant differences in intermolecular interactions and aggregation behavior. Such unique solution processibility allows for fabricating diverse continuous HOF membranes with high crystallinity and porosity through solution-casting approach on various substrates. Among them, HOF-BTB@AAO membranes show high $C_3H_6$ permeance ($1.979 \times 10^{-7}$ mol·s$^{-1}$·m$^{-2}$·Pa$^{-1}$) and excellent separation performance toward $C_3H_6$ and $C_3H_8$ (SF = 14). This continuous membrane presents a green, low-cost, and efficient separation technology with potential applications in petroleum cracking and purification.

Membrane-based gas separation technology, owing to its efficient energy utilization and ease of manipulation, has been one of the most promising alternatives to conventional energy-intensive separations[1–3]. Porous crystalline materials (PCMs) possess high surface area, well-defined pore sizes, and controllable functionality, making them excellent membrane candidates[4–6]. Although several studies have been reported to date[6–9], fabricating PCM membranes frequently encounters significant challenges in discontinuity, such as wrinkles, cracks, and aggregation, which severely affect their performance and practical applications[6]. Therefore, there is a need to explore new strategies for fabricating continuous membranes.

Hydrogen-bonded organic frameworks (HOFs), a type of porous crystalline material derived from organic building blocks self-assembled by hydrogen bonding and other weak interactions, not only inherit the merits of PCM, but also possess the distinctive quality of solution processability[10–15]. The latter allows for convenient processing and molding capacity via a solution-casting process. This process has the superiority of fabricating dense and continuous membranes through a facile, quick, sustainable, and low-cost approach. However, the aggregation state of HOF in the solution, whether as isolated molecules or small segments of assemblies, has yet to be clarified. This information is critically important to the microstructure and mesoscale morphology of obtained membranes. Furthermore, understanding the state of HOF in solution is a prerequisite for revealing the mechanism of HOF membrane growth, optimizing the solution process, and exploring new fabrication technology.

In this study, we investigated the aggregation state of a porous and crystalline HOF material (HOF-BTB) dispersed in a solvent using Cryo-electron microscopy (Cryo-EM), showing numerous small fragments with sizes ranging from 10 to 200 nm. Moreover, the Cryo-EM

[1]State Key Laboratory of Structural Chemistry, Fujian Institute of Research on the Structure of Matter, Chinese Academy of Sciences, 350002 Fuzhou, Fujian, P. R. China. [2]University of Chinese Academy of Sciences, 100049 Yuquan Road, Shijingshan District, Beijing, P. R. China. [3]Department of Chemistry, Faculty of Sciences, Tarbiat Modares University, P.O. Box 14115-175 Tehran, Iran. [4]These authors contributed equally: Qi Yin, Kuan Pang. ✉e-mail: tfliu@fjirsm.ac.cn

coupled with the three-dimensional electron diffraction (3D-ED) technique elucidated that the crystal structure of the fragments is identical to that of the bulky HOF obtained by single-crystal X-ray diffraction. Meanwhile, we confirmed the hydrogen-bonding interaction between BTB molecules in solution using 2D $^1$H-$^1$H nuclear Overhauser effect spectroscopy (NOESY). These findings provide valuable insights into the solution-processable nature of HOFs. Exploiting this nature, we fabricated a highly crystalline and continuous HOF membrane on macroporous anodic aluminum oxide (AAO) disks (designated as HOF-BTB@AAO, Fig. 1) and demonstrated the applicability of this method to other HOFs, diverse solvent, and different substrates. As a feedstock for polymer prouction, separation of propylene from propane is highly important but very challenging due to their similar molecular kinetic diameters ($C_3H_6$: 4.678 Å, $C_3H_8$: 4.3–5.118 Å) and boiling point ($C_3H_6$: −47.69 °C, $C_3H_8$: −42.13 °C)[16]. For HOF-BTB@AAO membrane, the well-defined pore structure, selective molecular sieving effect, and membrane continuity synergistically contributed to its higher permeance for $C_3H_6$ than $C_3H_8$ ($1.979 \times 10^{-7}$ vs $9.953 \times 10^{-9}$ mol·s$^{-1}$·m$^{-2}$·Pa$^{-1}$, respectively), leading to excellent separation performance of $C_3H_6$ from its mixture with $C_3H_8$. The work presented here provides an alternative of preparing crystalline HOF membrane with desired structure and high uniformity.

## Results

HOF-BTB is constructed from the triangular building block 1,3,5-Tris(4-carboxyphenyl)benzene (BTB), where each BTB ligand is connected to three adjacent BTBs via carboxyl-based hydrogen bonds (O−H...O) to form 2D honeycomb-like layers (Fig. 2, Supplementary Fig. 1)[17,18]. These layers interpenetrate into a complex 3D network, resulting in 1D undulated channels of ~10 Å. The successful synthesis of crystalline HOF-BTB was confirmed using powder X-ray diffraction (PXRD) patterns, $N_2$ sorption isotherms, and nonlocal density functional theory (NLDFT) pore size distributions (see Supplementary Figs. 2–4). It was interesting to find that dispersion of HOF-BTB in N,N-dimethyl formamidine (DMF) yielded a clear colloid solution with a noticeable Tyndall effect, which was absent in the solution of amorphous BTB ligands dissolving in DMF (see Fig. 3a, b). This phenomenon suggested that there may exist HOF-BTB nanoparticles in solution. The Ultraviolet−visible (UV-vis) spectra show that HOF-BTB colloid solution has a new adsorption band at 427 nm, which is absent in BTB

solution (Supplementary Fig. 5). A plausible reason is that the strong π−π interaction existed in HOF-BTB particle results in the decreased energy for π−π* transition and therefore red-shift adsorption of the colloid. To clarify our hypothesis, we performed additional analyses including 1D and 2D-NMR, zeta potential, and dynamic light scattering (DLS) measurements. The HOF-BTB colloid solution exhibited a higher Zeta potential (33.12 mV), therefore higher stability of colloid, compared to the BTB solution (15.65 mV) (see Supplementary Table 1)[19]. Moreover, DLS measurement showed that there are two kinds of nanoparticles with size about ~10 and ~137 nm in the freshly prepared HOF-BTB colloid (Supplementary Fig. 6a). This colloid gradually transformed to monodisperse with size ~100 nm after standing in ambient environment for 1 week (Supplementary Fig. 6b), and no precipitation can be observed in 1 month, indicating long-term stability. $^1$H-nuclear magnetic resonance (NMR) studies showed that the HOF-BTB colloid solution had a much broader peak width (13.1 ppm) than the BTB solution (Supplementary Fig. 7). A broader peak width often means a rapid chemical exchange rate[20]. Therefore, we can conclude that the carboxylate group in HOF-BTB shows a faster proton-exchange rate. This may be due to the O−H bonds involved in the complementary hydrogen bonds in HOF-BTB being slightly elongated relative to the free BTB molecules, resulting in a lower bond-dissociation energy and a faster proton exchange rate[21–23]. This hypothesis was further supported by the 2D $^1$H-$^1$H NOESY studies (Fig. 3a, b), which showed that a significant cross-peak at 13.1 and 3.3 ppm was observed for the BTB solution, whereas this peak was nearly absent in the HOF-BTB colloid solution. This difference may be due to the hydrogen-bonding interaction between $H_2O$ molecules and the carboxyl groups of free ligand molecules in BTB solution, which is replaced by the complementary hydrogen bonds between two carboxyl groups in the HOF-BTB colloid solution.

To gain structure information of these specimens, we rapidly froze HOF-BTB colloid in liquid nitrogen to maintain their state in solution and then analyzed using Cryo-EM. The transmission electron microscope (TEM) image showed the presence of nanosized fragments ranging from 10 to 150 nm (Fig. 3c), which is in line with the particle sizes deduced from DLS measurement. The lattice stripes can be clearly observed in Fig. 3d, confirming the high crystallinity of particles in colloid. In order to further establish the precise structure of nanoparticles, 3D-ED technology was employed to obtain its

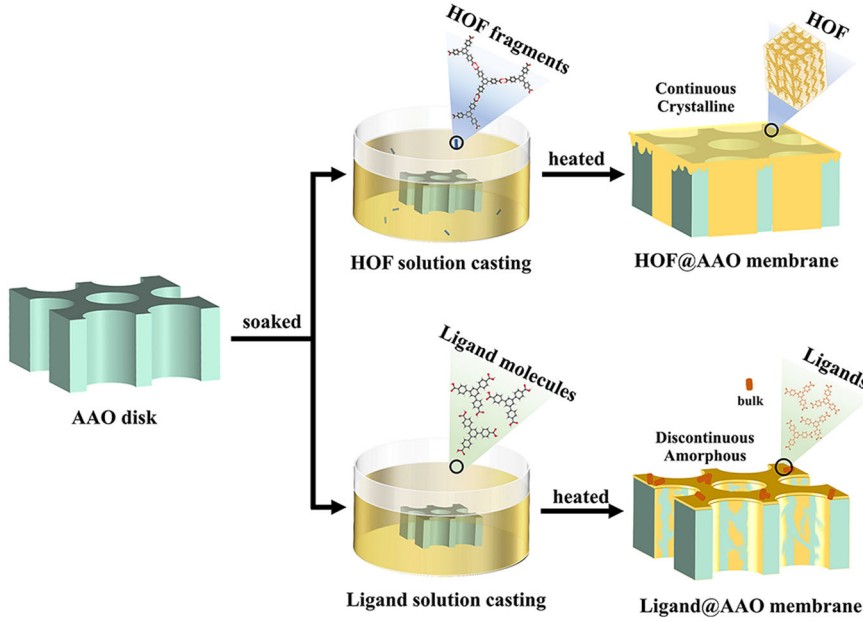

**Fig. 1 | Schematic diagram of the preparation of HOF@AAO membrane through a solution-casting approach.**

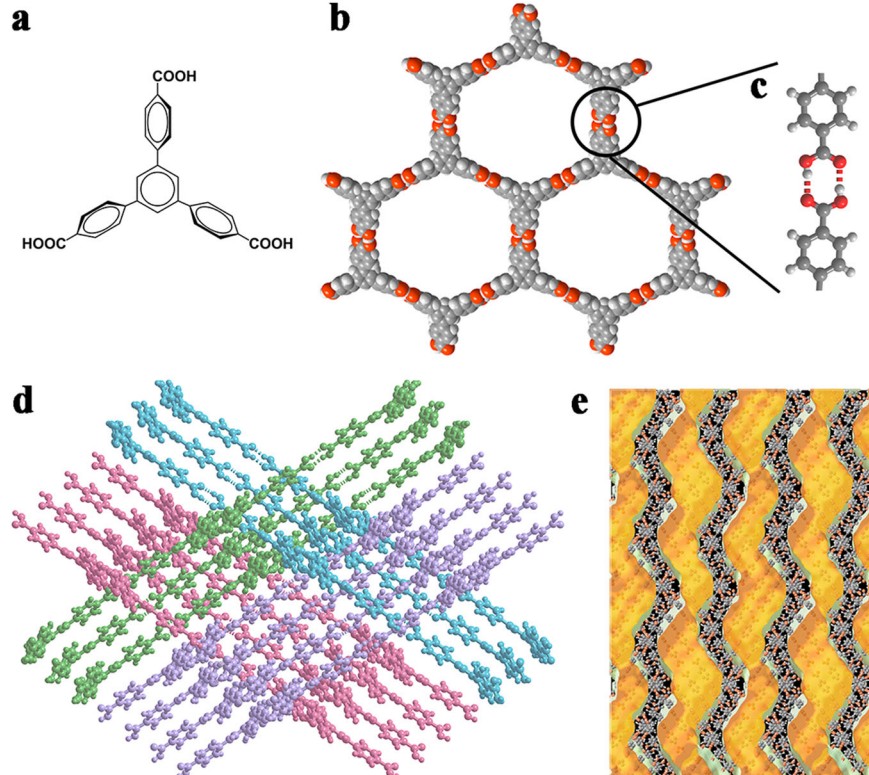

**Fig. 2 | Crystal structure of HOF-BTB. a** Chemical structure of BTB ligand. **b** The honeycomb-like 2D layer interconnected through **c** complementary hydrogen bonding. **d** The interpenetrated network and **e** Internal pore surface mapping of HOF-BTB (Yellow: the void space in structure).

crystallographic unit cell. As shown in Fig. 3e and Supplementary Fig. 8, the obtained unit cell agrees with that of the bulky HOF determined by single-crystal X-ray diffraction (SCXRD, Supplementary Table 2). These findings demonstrate that HOF material can be dispersed in organic solvent to produce a colloid solution, where supramolecular assemblies fragment into small particles while retaining their original porosity and crystallinity. Although the solution processability feature of some supramolecular assemblies have been mentioned by some studies[24], our work visualize their morphologies and confirm the intact assemblies of HOF in solution.

The maintained crystallinity of HOF-BTB in the solution offered the opportunity to fabricate continuous membranes through a facile solution-casting approach. To this end, an AAO disk with highly-ordered nanochannel array was selected as a porous template to geometrically confine HOF-BTB inside the pore. Because of the excellent affinity to the AAO disk (contact angle of 13.9°, Supplementary Fig. 9), DMF solvent was chosen to disperse HOF-BTB for a casting solution. Therewith, the AAO disk was immersed in HOF-BTB casting solution at room temperature (RT) and then heated at 100 °C. This casting operation was repeated several times to get HOF-BTB@AAO membrane (see the detailed procedure in supplementary information 2.3). During this process, different HOF-BTB concentrations (2, 5, and 7 mg/mL), and the number of repeated casting operation (2, 5, and 10 cycles) were screened to achieve optimal crystallinity and uniformity. Top view scanning electron microscopy (SEM) images revealed that using 7 mg/mL HOF-BTB solution for 5-cycle casting operations led to the most even and continuous HOF casting on the AAO surface (Supplementary Figs. 10 and 11), and the HOF-BTB layer is about 126 nm as observed in the cross-sectional SEM image (Supplementary Figs. 12 and 13). Gas permeance experiments also confirmed this condition generating a membrane with best gas separation performance (Supplementary Figs. 14 and 15). We speculated that as the number of solution-casting cycles increased, defects may be repaired during the hydrogen-bond

assembling processes, whereas excessive number of casting cycles would lead to partial dissolution of the crystallized HOF-BTB, resulting in more defects in the coating layer and AAO channel. The cross-sectional SEM images showed that compared with the bare AAO disk, HOF-BTB@AAO exhibited a lower contrast between the channel surface and the cross section of the AAO wall. The presence of organic components would backscatter more electrons and appear brighter than the bare AAO (Fig. 4a, b insets). Furthermore, SEM-EDS mapping showed a much higher C content and lower Al content in HOF-BTB@AAO than that in the AAO disk (Supplementary Table 3), conforming the success of the casting process. Besides, in the Fourier Transform Infrared (FTIR) spectrum of HOF-BTB@AAO membrane, the appearance of BTB's characteristic bands at 1687 cm$^{-1}$ (C=O stretching vibration) and 767 cm$^{-1}$ (aromatic group) illustrated the presence of BTB in the membrane (Supplementary Fig. 16)[25]. In addition, the high $CO_2$ uptake of HOF-BTB@AAO over the AAO disk at 298 K confirmed the porosity of the composite membrane (Supplementary Fig. 17). As a comparison, amorphous BTB ligand was dissolved in DMF to produce membranes via the same approach instead of using crystalline HOF-BTB colloid. The resulting membrane did not exhibit the continuous coating on the AAO surface, and its PXRD pattern did not show any peaks (Figs. 4c and S1). These results indicate that the solute's crystallinity plays a critical role in the solution-casting process, being a prerequisite for a uniform and highly-crystalline HOF membrane.

To further confirm the applicability of this approach to other substrates, HBG, ITO glass, PET-ITO, and Cu sheet were selected to fabricate composite membranes with the same solution-casting operation, which gave rise to continuous crystalline HOF-BTB@HBG, HOF-BTB@ITO, HOF-BTB@PET-ITO, and HOF-BTB@Cu membranes as shown in Fig. 4d–k, Supplementary Fig. 18. The applicability of this approach to different HOF materials was also illustrated by using other reported HOFs (PFC-1 and PFC-72-Co) constructed from 1,3,6,8-tetrakis(p-benzoic acid)pyrene (TBAPy) and [5,10,15,20-Tetrakis(4-

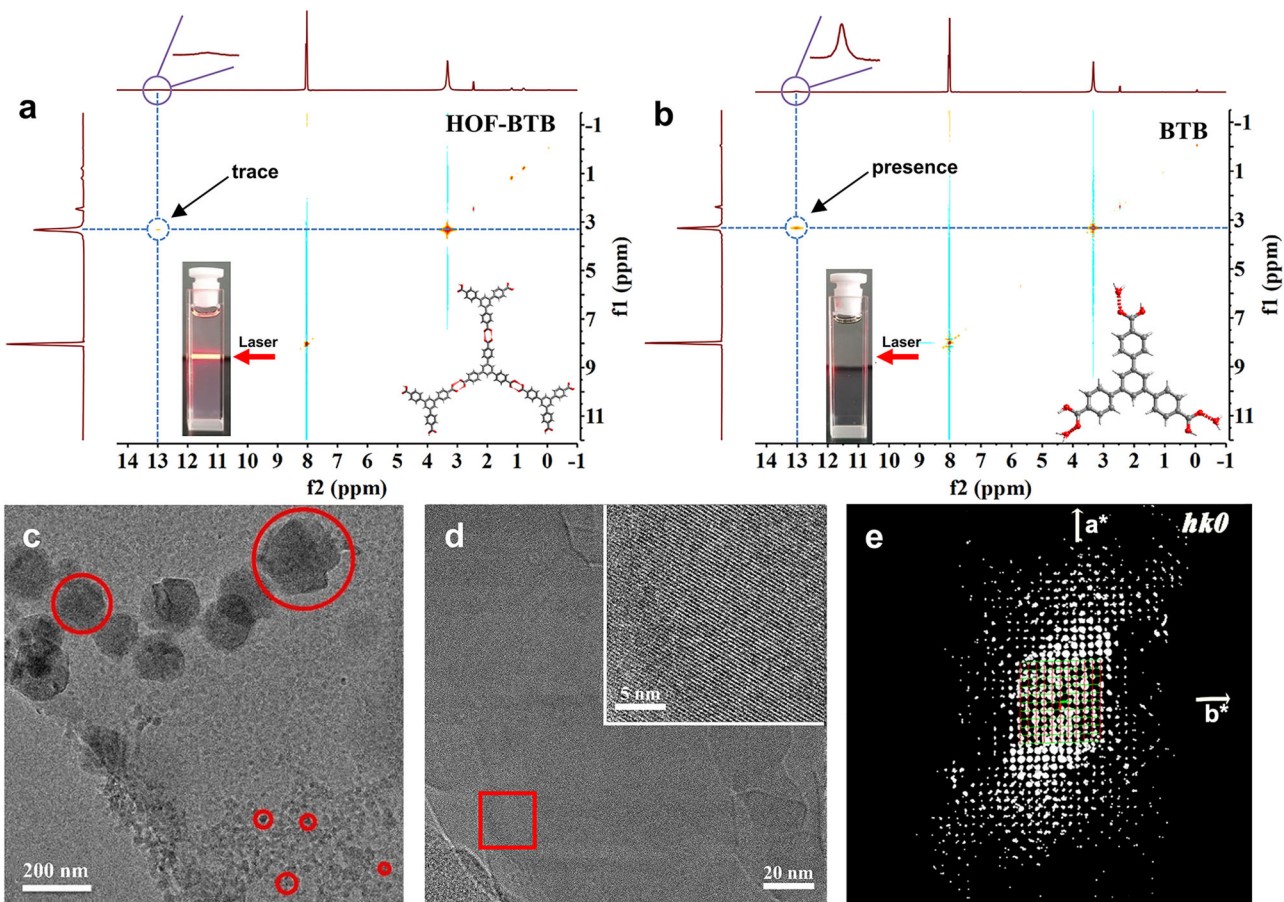

**Fig. 3 | Characterization of HOF solution.** 2D $^1$H-$^1$H NOESY spectra of **a** 5 mg HOF-BTB and **b** 5 mg amorphous BTB ligands in 560 μL DMSO-$d_6$ solution. Tyndall effect of the solution with 5 mg/mL HOF-BTB (**a** inset) and amorphous BTB ligands (**b** inset) in DMF. **c**, **d** Cryo-transmission electron microscopy (Cryo-EM) images of 5 mg/mL HOF-BTB in DMF solution. Inset: Magnified TEM image of the red-boxed region. **e** Projection of a reconstructed 3D reciprocal lattice along c* direction.

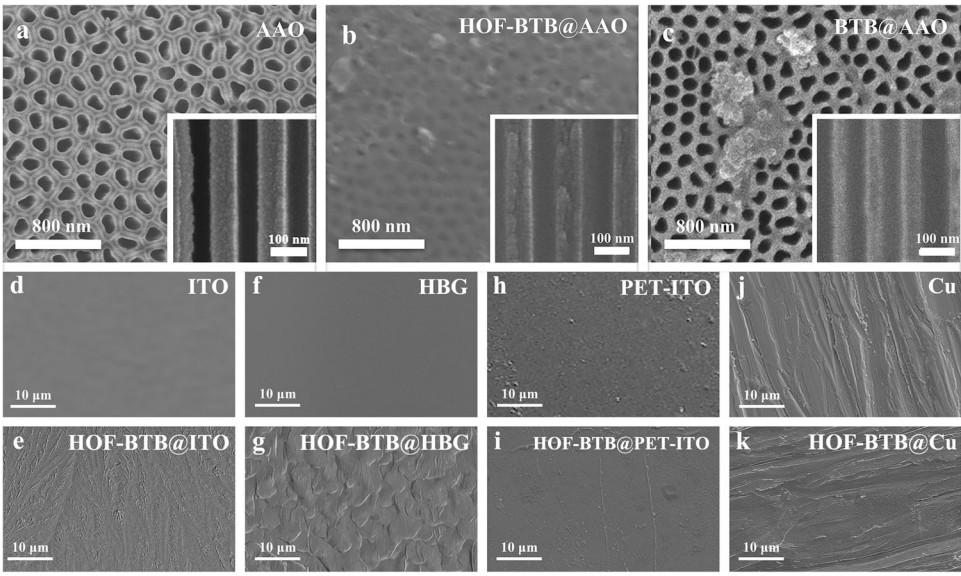

**Fig. 4 | SEM images of membranes.** Top-view SEM images of **a** barely AAO disk, **b** HOF-BTB@AAO membrane, and **c** BTB@AAO membrane, insets are their corresponding cross-section images. SEM images of **d** ITO, **e** HOF-BTB@ITO, **f** HBG, **g** HOF-BTB@HBG, **h** PET-ITO, **i** HOF-BTB@ PET-ITO, **j** Cu sheet, and **k** HOF-BTB@Cu membranes.

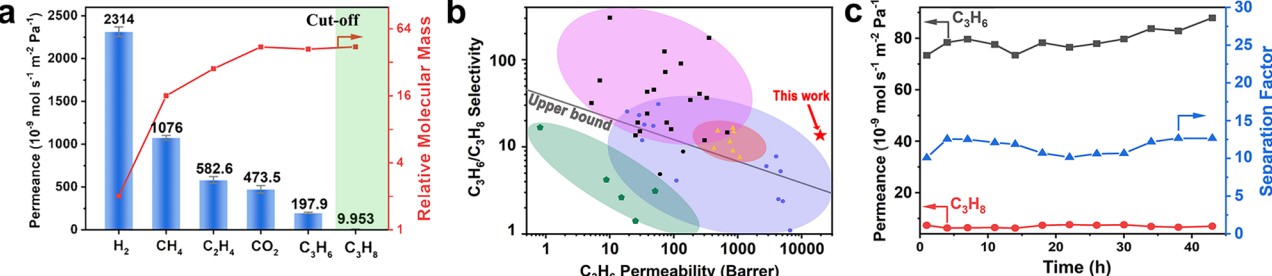

**Fig. 5 | Separation performance of HOF-BTB@AAO membrane. a** Single gas permeance of HOF-BTB@AAO membrane as a function of the relative molecular mass. Error bars represent standard deviations. **b** Comparison of $C_3H_6/C_3H_8$ separation performance with other representative membranes, such as Zeolitic Imidazolate Framework-8 membranes (square), mixed matrix membranes (rotundity), Facilitated Transport Membranes (triangle), and polymers membranes (pentagon). **c** Long-term durability of HOF-BTB@AAO membrane for $C_3H_6/C_3H_8$ separation at RT and 100 kPa.

carboxyphenyl)porphyrinato]-Co(II) (TCPP-Co), respectively (Supplementary Figs. 19 and 20)[26]. The powdery PFC-1 and PFC-72-Co can be dispersed in DMF to form a clear casting solution with visible Tyndall effect (Supplementary Fig. 21) and then generated continuous crystalline HOF@AAO membranes using the same solution-casting technique (Supplementary Figs. 22 and 23). In contrast, amorphous TBAPy ligand dissolving in DMF failed to produce a crystalline and continuous membrane via the same procedure (Supplementary Fig. 22a). Besides, the HOF colloid can also be prepared using other solvents as demonstrated by dispersing HOF-BTB in N,N-dimethyl aniline (DMA) for fabricating crystalline HOF membrane (named HOF-BTB@AAO_DMA, Supplementary Fig. 22c). In short, these experiments confirmed the generality of this solution-casting approach.

In view of the inherent porosity and undulated channels of HOF-BTB, gas separation performance of HOF-BTB@AAO membrane was evaluated. The bare AAO disk showed ultra-high gas permeance (out of detection limit) for all the studied gases, and its small gas separation selectivity had been reported[27], thereby ruling out the separation effect from the AAO support. HOF-BTB@AAO membrane showed single gas permeance of $2.314 \times 10^{-6}\,mol\cdot s^{-1}\cdot m^{-2}\cdot Pa^{-1}$ for $H_2$, $1.076 \times 10^{-6}\,mol\cdot s^{-1}\cdot m^{-2}\cdot Pa^{-1}$ for $CH_4$, $5.826 \times 10^{-7}\,mol\cdot s^{-1}\cdot m^{-2}\cdot Pa^{-1}$ for $C_2H_4$, $4.735 \times 10^{-7}\,mol\cdot s^{-1}\cdot m^{-2}\cdot Pa^{-1}$ for $CO_2$, $1.979 \times 10^{-7}\,mol\cdot s^{-1}\cdot m^{-2}\cdot Pa^{-1}$ for $C_3H_6$, and $9.953 \times 10^{-9}\,mol\cdot s^{-1}\cdot m^{-2}\cdot Pa^{-1}$ for $C_3H_8$ at 298 K and 1 bar (Fig. 5a), and the permeance ratios of $H_2$, $CH_4$, $C_2H_4$, $CO_2$, $C_3H_6$, and $C_3H_8$ equal to 4.89:2.27:1.23:1:0.4:0.02. Their Knudsen selectivity, calculated by taking the inverse of the square root of the molecular mass, were determined to be 4.69:1.66:1.11:1:1.02:1. For the first five molecules, their permeance ratios are quite close to their Knudsen selectivities, indicating that the main transport mechanism is Knudsen diffusion, rendering such a membrane separation technology unattractive practical applications[28]. However, a distinct cutoff permeance between $C_3H_8$ and other gas molecules can be observed, suggesting a molecular sieving diffusion mechanism for gas separation and the promising potential for propane-related separation.

For HOF-BTB@AAO membrane, the ideal separation factor (ISF) of propylene/propane is 20 (i.e., 0.4:0.02), which is much higher than the corresponding Knudsen selectivity (1.00), demonstrating promising potential in propylene/propane separation. It is noticeable that the propylene/propane separation performance of HOF-BTB@AAO exceeded the single gas upper bound line, outperforming the majority of reported membranes (Fig. 5b). Moreover, three repeated measurements show small standard deviations, confirming the high reproducibility of this protocol.

Aiming for practical application, we further evaluated the binary $C_3H_6/C_3H_8$ separation performances of HOF-BTB@AAO membrane at different conditions. As shown in Supplementary Fig. 24, the mixed separation factor (SF) for the binary equimolar $C_3H_6/C_3H_8$ mixture reached 14 at 27 °C and 100 kPa, which approximated the prediction based on ISF, and this separation capacity was maintained during the

long-term experiment of 44 h (Fig. 5c). The analyses of temperature dependence to the permeances of both $C_3H_6$ and $C_3H_8$ revealed the enhanced permeance as temperature increases. Furthermore, the activation enthalpies of permeation ($E_p$) were calculated to be 9.3 and 13.3 kJ/mol for $C_3H_6$ and $C_3H_8$, respectively (Supplementary Equation 9, Supplementary Fig. 25); whereas, the heat of sorption values (ΔH) for $C_3H_6$ and $C_3H_8$ were almost the same (~−26.6 kJ/mol, Supplementary Figs. 26 and 27). The resultant diffusional activation energies ($E_d$) of $C_3H_6$ and $C_3H_8$ were 35.9 and 39.9 kJ/mol, respectively (Supplementary Equation 11). This distinct difference in $E_d$ for two gases strongly supports that the effective $C_3H_6/C_3H_8$ separation performance of this membrane belong to the molecular sieving diffusion principle[29]. Simultaneously, the effect of feed flow ratio and pressure on the separation performance were investigated. As shown in Supplementary Figs. 24 and 28, although the SF for $C_3H_6/C_3H_8$ mixture gradually decrease along with the increase of total pressure and $C_3H_8$ percentage of feed flow, $C_3H_8$ remained very low permeance throughout the process, confirming the excellent molecular sieving effect toward $C_3H_8$. Overall, the HOF-BTB@AAO membrane manifested desired robustness, high selectivity, and long-term stability toward $C_3H_6/C_3H_8$ separation.

## Discussion

In summary, we present the preservation of the original assembled porous structure of HOF after being dispersed in organic solvent, and its crystallographic structure can be determined through in-situ Cryo-EM coupled with 3D-ED studies. This unique solution processability enables the preparation of various continuous HOF membranes with high crystallinity through the facile and effective solution-casting approach, which was illustrated by the highly-crystalline and uniform HOF-BTB@AAO, HOF-BTB@HBG, HOF-BTB@ITO, HOF-BTB@PET-ITO, HOF-BTB@Cu, HOF-BTB@AAO_DMA, PFC-1@AAO and PFC-72-Co@AAO membranes. Among these membranes, HOF-BTB@AAO membrane emerged with high $C_3H_6$ permeance ($1.979 \times 10^{-7}\,mol\cdot s^{-1}\cdot m^{-2}\cdot Pa^{-1}$) and effective separation performance of $C_3H_6$ over $C_3H_8$ (SF = 14) at 1 bar and room temperature due to the molecular sieving effect, demonstrating its potential as an energy-saving separation technology for pratical applications.

## Data availability

All data generated or analyzed during this study are included in this published article and its supplementary information files. Source data are provided with this paper.

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

## Acknowledgements

The authors thank Prof. Dong Wang and Xiao-Rui Ren from Institute of Chemistry, Chinese Academy of Sciences for the help on Cryo-EM experiments. The authors thank National Natural Science Foundation of China (NSFC, Grant No. 22071246), Chinese Academy of Sciences (CAS) youth interdisciplinary team (Grant No JCTD-2022-12), CAS-Iranian Vice Presidency for Science and Technology Joint Research Project (121835KYSB20200034), Iran National Science Foundation (99009822).

## Author contributions

All authors have given approval to the final version of the manuscript. Prof. T.F.L. guided the project. Dr. Q.Y. and K.P. carried out most of the synthesis, characterization, and data analysis in this work. Dr. Q.Y. carried out the original draft & editing. Prof. L.H. assisted Cryo-EM analysis. Y.N.F. and X.Y.L. assisted SEM image collection. Prof. T.F.L. and Prof. A.M. assisted funding acquisition and revised the manuscript.

## Competing interests

The authors declare no competing interests.
