## [Peer Review File · Nature Communications]

Hydrogen-bonded Organic Frameworks in Solution: Toward Continuous and High-crystalline Membranes for Gas SeparationReviewers' Comments:

Reviewer #1:

Remarks to the Author:

The authors here investigated the behaviour of HOFs in solution. Techniques including NMR, zeta potential, cryo-electron microscopy and 3d electron diffraction have been used to understand the behaviours of re-dissolved HOFs, especially when compared with that of the solution of the building units, discrete molecules. Furthermore, the authors have demonstrated the fabrication of HOF membranes using the re-dissolved HOFs solutions. One of the membrane systems was claimed to be able separate C₃H₆ from C₃H₈ efficiently. Although neither the structures nor the membrane fabrication method is new, the work provides some new observation of the HOFs in solution using some state-in-art techniques, which could benefit the fabrication of HOFs membranes in the future. However, this work will need major improvement before meeting criteria for publishing in Nat. Comm. Here are questions and comments in details:

1. This work using a similar method to prepare HOFs membranes as a previous work (Angew. Chem. Int. Ed. 2020, 59, 3840 –3845), in which the HOFs membranes form through nucleation induced crystallization mechanism. In fact, the results reported in this paper has also support such membrane formation mechanism. A slight modification of the method is the authors here sonicated the HOFs solutions for a few minutes before casting. Have the authors checked the state of the sonicated HOFs solutions, as one would expect that sonication would facilitate the dissolution of HOFs crystals and reduce crystal nucleus?
2. One of the highlights of this work is to investigate the HOFs structures in solution, especially the authors used two state-in-art techniques: cryo-electron microscopy and 3D electron diffraction. However, both techniques involve a rapid-freezing process which may promote the crystallization of the HOFs structure, which may not best represent the states of the HOFs in solution. Some more common techniques that directly works on solutions maybe able to provide more direct information: such as UV-vis, DLS...
3. The authors compared using two types of HOFs solution to form membranes: re-dissolved HOFs and solution of discrete ligands. It is useful to use these comparisons to prove the necessity of have HOFs crystals in the solution in order to form continuous membranes. However, it would be more interesting to optimize a condition that can fabricate continuous membranes directly from discrete HOF ligands. For example: would it be possible to allow the discrete ligands solution to stand/evaporate in order to form crystals nucleus before casting. Or simply increase the concentration of the ligands in the solution.
4. The authors using DMF as solvents for the characterization of the solution state as well as for the fabricating of membranes, have there any other solvents been studied? It will be interesting to understand the role of solvents play in the forming of membranes. And for different ligands, will the solvents have different influence? Furthermore, if the authors want to claim the method is a general method for fabricating HOFs membranes, a couple of more examples besides BTB and PFC-1 will be beneficial.
5. Could the authors explain why there is significant drop of adsorption of CO₂ when HOF-BTB formed membrane in S14?

Reviewer #2:

Remarks to the Author:

The authors used the solution-casting method to fabricate crystalline HOF membranes on different substrates. The resulting HOF@AAO membrane exhibited the capability to separate the C₃H₆/C₃H₈ mixture. The topic is intriguing; nevertheless, there are several instances of inaccurate data processing, which need to be corrected/clarified before publication.

1. Please reorganize the layout of the Supporting Figures, as the current sequence is unclear. For instance, Figure S11 is followed by Figure S20, and Figure S3 is followed by Figure S18. Additionally, there seems to be confusion regarding the labeling of Figures 3e and f. In the text, Figures 3e and f

are used to reference PFC-1@AAO and TBAPy@AAO, respectively, but in the figures, they are labeled as HOF-BTB@ITO and HBG.

2. LINE 216-223 should be moved to the introduction section.

3. Most of the experimental PXRD patterns have poor matching with the simulated ones. In addition, no crystallinity can be found in Figure S3c.

4. In FTIR (Figure S13), there is a red-shift of HOF-BTB at around 1689 cm^{-1} ; is it caused by hydrogen bonding? Why is there no shift for the HOF-BTB@AAO membrane?

5. The gas separation results need to be well organized. (1) In Figure 4b, the x-axis is C_3H_6 permeability, which needs a clear membrane thickness for calculation; however, there is no such information on membrane thickness. For supported membranes, permeance is a more common and accurate expression. (2) Figure S23, the testing pressure is below 1 bar, how is it possible? If the pressure difference is the driving force, it should be clearly marked.

6. For membrane fabrication, could the author explain why 5-cycle leads to a continuous membrane, but 10-cycle could not? If the "dissolution of the crystallized HOF-BTB" is the reason, there should be a continuous membrane with defects, but we can only find some particles like the 2-cycle sample.

7. Cross-sectional SEM images are needed to show the thickness of the HOF membrane, where AAO support thickness should not be included.

Reviewer #3:

Remarks to the Author:

The authors presented a method to produce HOF membranes on various types of substrates. They showed the membrane has nice separation performance for C_3H_6 and C_3H_8 . However, some issues need to be resolved properly before it can be considered for acceptance in Nature Communication.

Introduction

1. The authors emphasized that the study is about HOF in solution state, which is quite misleading. They are HOF nanoparticles. In many cases, these nanoparticles tend to aggregate and cannot form stable colloid in solution. Rather than write something like solution state, it would be better to say forming stable colloid in solution and study the properties of the colloid.

2. The authors claimed that they use 3DED technique to determine the crystal structure. However, I didn't find any data statistics regarding the structure determination process, such as R values, data quality, etc. Only unit cell parameters were provided. The space group is not presented at all.

Results

3. "The larger particle size and improved colloid stability of HOF-BTB over amorphous BTB" BTB dissolved in solution is not amorphous. Amorphous is in solid state, not in solution.

4. "there may be segments of supramolecular assemblies in HOF-BTB solution"

Do you mean some clusters of HOF-BTB or some precursor of HOF-BTB? Could you estimate how large are the clusters or segments and why they can stabilize the colloid?

5. "This difference may be due to the interaction between H_2O molecules and the carboxyl groups of free BTB molecules in solution"

What kind of interaction does the authors refer in here? Hydrogen bonding interaction? Also where does H_2O molecules comes from? The system is in DMSO solution.

6. "(HRTEM) image observes the spatial frequencies characteristic of $(0\ 7\ 7)$, $(0\ 7\ \bar{7})$, $(0\ \bar{7}\ 7)$, $(0\ \bar{7}\ \bar{7})$, $(0\ 0\ (14))$, and $(0\ 0\ 14)$ planes of the HOF-BTB structure"

The indexing is wrong. There is no such systematic absences in crystallography. Either the zone axis is wrong or the unit cell is incorrect. Please take an electron diffraction pattern in the same orientation.

7. In Figure 2 (e), xyz axis is not very useful in reciprocal space. Please use a^* , b^* and c^* axis

8. Figure s3 d has some problems of overlapping text.

9. Add a, b and c axis in the figure S17

10. Figure 2(e) clearly shows the angle between a^* and b^* is not 90 degree. However, in Figure S7, all the angles become around 90 degree. Also, the 3D reciprocal space in Figure S7 is not from a single crystal. Please determine the correct unit cell. Also provide the systematic absence to determine the space group

11. Please add the crystallography data statistics table.
 12. Please attach the cif file and upload it to CCDC database and provide the corresponding CCDC number.
 13. Figure S3 showed the PXRD patterns of HOF on various types of substrates. All of them showed very poor crystallinity. For HOF on PET-ITO, the small peaks between 10 and 20 degree are not HOF-BTB. I suggest to remove this. PXRD for membrane on other substrate showed very few peaks and they are very wide. I am not even sure if they are HOF-BTB or not. Probably on HBG might be HOF-BTB. But on ITO and Cu, only the first peak matches and only one peak is available. So you cannot say it is not HOF-BTB but it would be quite uncertain to say it is HOF-BTB.
 14. Why HOF membrane can have good separation performance for C₃H₆/C₃H₈? Are there any explanations for the separation mechanism?
 15. Please obtain the size distribution of HOF-BTB nanoparticles before and after the dispersion process.
- Conclusion
16. Again dissolve here is misleading. HOF-BTB are nanoparticles and able to disperse well in organic solvent.

REVIEWER COMMENTS

Reviewer #1 (Remarks to the Author):

The authors here investigated the behaviour of HOFs in solution. Techniques including NMR, zeta potential, cryo-electron microscopy and 3d electron diffraction have been used to understand the behaviours of re-dissolved HOFs, especially when compared with that of the solution of the building units, discrete molecules. Furthermore, the authors have demonstrated the fabrication of HOF membranes using the re-dissolved HOFs solutions. One of the membrane systems was claimed to be able separate C₃H₆ from C₃H₈ efficiently. Although neither the structures nor the membrane fabrication method is new, the work provides some new observation of the HOFs in solution using some state-in-art techniques, which could benefit the fabrication of HOFs membranes in the future. However, this work will need major improvement before meeting criteria for publishing in Nat. Comm. Here are questions and comments in details:

1. This work using a similar method to prepare HOFs membranes as a previous work (Angew. Chem. Int. Ed. 2020, 59, 3840–3845), in which the HOFs membranes form through nucleation induced crystallization mechanism. In fact, the results reported in this paper has also support such membrane formation mechanism. A slight modification of the method is the authors here sonicated the HOFs solutions for a few minutes before casting. Have the authors checked the state of the sonicated HOFs solutions, as one would expect that sonication would facilitate the dissolution of HOFs crystals and reduce crystal nucleus?

Response: In the mentioned research, the authors dissolved HOF in DMSO solvent, and the obtained solution was not subject to any characterizations. Then they directly coated this solution onto a substrate. The state of matter in solution, whether as isolated molecules or small segments of assemblies, was not clear in this case. Consequently, it's uncertain whether the resulting crystalline membrane was formed through a crystal nucleation-growth process or a growth process based on existing crystal nuclei. In this work, differently, we focus on elucidating the state of HOFs in solution and how its aggregation state affects the membrane continuity and morphology. A range of techniques, including NMR, zeta potential analysis, Cryo-electron microscopy, and 3D electron diffraction, were employed to characterize the HOF colloid solution.

The sonication procedure applied here is of short duration (≤ 5 s) to quickly dissolve HOF in DMF. This can also be achieved by gently shaking the vial for about 20s (with hand). All these operations gave rise to a solution with distinct Tyndall effect that is absent in the discreet BTB ligand solution (Figure R1). Subsequently, this HOF solution sample was subject to the subsequent Cryo-electron microscopy (Cryo-EM) and Dynamic Light Scattering (DLS) measurement, and results showed that there exist many crystalline fragments in the solution with the size ranging from 10 nm to 200 nm (Figure R2). Considering the very low energy of manually shaking, we believe it's a dissolution process rather than mechanically breaking crystals into nanoparticles. As a result, the process (sonicated or manually shaking) does not destroy the building block self-assembly, and the maintained crystal nuclei facilitates the later membrane fabrication. We have added more detailed description and discussion on the sonication procedure in the revised supporting information and the editings are highlighted in yellow.

Figure R1. Optical photographs of DMF solution with a) 5 mg/mL BTB ligands and d) 5 mg/mL HOF-BTB illuminated by a 635 nm red laser pointer. Figure R1b shows apparent Tyndall effect.

Figure R2. a) Cryo-Transmission electron microscopy (Cryo-EM) image and b) Dynamic Light Scattering (DLS) derived particle size distribution of 5 mg/mL HOF-BTB in freshly prepared DMF solution (the experiments were repeated twice to ensure the reliability of results).

2. One of the highlights of this work is to investigate the HOFs structures in solution, especially the authors used two state-in-art techniques: cryo-electron microscopy and 3D electron diffraction. However, both techniques involve a rapid-freezing process which may promote the crystallization of the HOFs structure, which may not best represent the states of the HOFs in solution. Some more common techniques that directly works on solutions maybe able to provide more direct information: such as UV-vis, DLS...

Response: We thank this reviewer for his/her constructive suggestion. The UV-vis spectrum and DLS measurement were conducted (Figure R3-4), and the results and related discussion were added into the revised manuscript and supporting information.

The Ultraviolet-visible (UV-vis) spectra show that HOF-BTB-based solution shows a new adsorption in 427 nm, which is absent in the discrete BTB solution (Figure R3). A plausible reason is that the strong π - π interaction existed in HOF-BTB particle results in the decreased energy for π - π^* transition and therefore red-shift adsorption of the colloid. The Dynamic Light Scattering (DLS) measurement shows that there are two kind of nanoparticles with size of ~ 10 nm and ~ 137 nm in HOF-BTB colloid solution (Figure R4), which is in line with the information observed in TEM image.

Figure R3. Ultraviolet–visible (UV-vis) spectra of HOF-BTB and BTB ligand in DMF solution, respectively.

Figure R4. Dynamic light scattering (DLS) measurement conducted on a freshly prepared solution of HOF-BTB in DMF (the experiments were repeated twice to ensure the reliability of results).

3. The authors compared using two types of HOFs solution to form membranes: re-dissolved HOFs and solution of discrete ligands. It is useful to use these comparisons to prove the necessity of have HOFs crystals in the solution in order to form continuous membranes. However, it would be more interesting to optimize a condition that can fabricate continuous membranes directly from discrete HOF ligands. For example: would it be possible to allow the discrete ligands solution to stand/evaporate in order to form crystals nucleus before casting. Or simply increase the concentration of the ligands in the solution.

Response: Thank this reviewer for the constructive suggestion. It's indeed more intriguing if one can fabricate continuous membranes through one-step synthetic strategy from discrete ligands. In fact, we have been striving to achieve this goal for nearly every building block synthesized in our lab, but only very few of them can achieve crystalline and continuous HOF membrane through the one-step solution-process approach. There is a pressing need to explore more efficient strategies for membrane fabrications. Our work highlights a promising avenue in which the maintaining HOF fragments in solution act as seed crystals and increase the chance of successfully preparing the desired HOF membrane compared to the traditional one-step process. Therefore, we provide an alternative for fabricating HOF membrane with desired structure.

4. The authors using DMF as solvents for the characterization of the solution state as well as for the fabricating of membranes, have there any other solvents been studied? It will be interesting to understand the role of solvents play in the forming of membranes. And for different ligands, will the solvents have different influence? Furthermore, if the authors want to claim the method is a general method for fabricating HOFs membranes, a couple of more examples besides BTB and

PFC-1 will be beneficial.

Response: Thank this reviewer for this constructive suggestion. For the fabrication of HOF-BTB membrane, besides DMF, we can also use N, N-Dimethylaniline (DMA) for preparing HOF colloid solution, and it gave us a similar crystalline HOF membrane (named HOF-BTB@AAO_DMA). As to the role of solvents in the formation of membrane, we found protic solvent such as DMA and DMF are usually capable of yielding well-dispersed HOF colloid solution. A plausible explanation is that the high polar protic solvent might compete with the existing hydrogen bonds in network, reduce the particle size, and finally reach a new equilibrium.

To address the question on the universality of this method, we prepared another HOF (PFC-72-Co) and some experiments (crystal structure, PXRD, N₂ isotherms, and NLDFT pore distribution) were used to characterize its crystallinity and purity (Figure R5-6). Then we used DMF to dissolve PFC-72-Co for fabricating HOF membrane (named PFC-72-Co@AAO). The obtained solution also showed noticeable Tyndall effect, which was absent in the solution of TCPP-Co ligand in DMF (Figure R7). The successful preparation of crystalline TCPP-Co membrane was verified by the PXRD patterns (Figure R8). These experimental results have been added into revised manuscript and supporting information, and the editings were highlighted in yellow.

Figure R5. Crystal structure of PFC-72-Co. a) The structure of building block; b) the connection of adjacent building blocks; c-e) The packing structure seeing from different directions with highlighting the geometry and surface of channel.

Figure R6. a) PXRD patterns, b) N₂ isotherms at 77 K, and c) NLDFT pore size distribution of powdery PFC-72-Co.

Figure R7. Optical photographs of DMF solution with a) 5 mg/mL TCPP-Co ligands, and b) 5 mg/mL PFC-72-Co illuminated by a 635 nm red laser pointer.

Figure R8. PXRD patterns of a) simulated HOF-BTB, bare AAO disk, and HOF-BTB@AAO_DMA membranes, b) simulated PFC-72-Co, bare AAO disk, and PFC-72-Co@AAO membranes.

5. Could the authors explain why there is significant drop of adsorption of CO₂ when HOF-BTB formed membrane in S14?

Response: The HOF-BTB@AAO membrane is composed of HOF-BTB and AAO. The macroporous AAO accounts for the main component of the entire material and has very low CO₂ adsorption capacity. Thus, the CO₂ adsorption of the membrane exhibits a substantial decrease compared to that of powdery HOF-BTB.

Reviewer #2 (Remarks to the Author):

The authors used the solution-casting method to fabricate crystalline HOF membranes on different substrates. The resulting HOF@AAO membrane exhibited the capability to separate the C₃H₆/C₃H₈ mixture. The topic is intriguing; nevertheless, there are several instances of inaccurate data processing, which need to be corrected/clarified before publication.

1. Please reorganize the layout of the Supporting Figures, as the current sequence is unclear. For instance, Figure S11 is followed by Figure S20, and Figure S3 is followed by Figure S18. Additionally, there seems to be confusion regarding the labeling of Figures 3e and f. In the text, Figures 3e and f are used to reference PFC-1@AAO and TBAPy@AAO, respectively, but in the

figures, they are labeled as HOF-BTB@ITO and HBG.

Response: We thank this reviewer for the careful reading and pointing this out. These labels have been revised accordingly.

2. LINE 216-223 should be moved to the introduction section.

Response: The mentioned part has been revised accordingly.

3. Most of the experimental PXRD patterns have poor matching with the simulated ones. In addition, no crystallinity can be found in Figure S3c.

Response: After optimizing the synthetic condition, we now get membranes with higher crystallinity and its PXRD pattern are shown in Figure R9 in the revised supporting information.

Figure R9. PXRD patterns of a) simulated HOF-BTB, ITO glass, and HOF-BTB@ITO membrane, b) simulated HOF-BTB, HBG glass, and HOF-BTB@HBG membrane, c) simulated HOF-BTB, ITO-PET substrate, and HOF-BTB@ITO-PET membrane, d) simulated HOF-BTB, Cu sheet, and HOF-BTB@Cu membrane.

4. In FTIR (Figure S13), there is a red-shift of HOF-BTB at around 1689 cm^{-1} ; is it caused by hydrogen bonding? Why is there no shift for the HOF-BTB@AAO membrane?

Response: We agree with this reviewer's idea that the red-shift of HOF-BTB might be caused by hydrogen bonding. According to previous studies, a carboxylic acid involving in hydrogen bonding usually exhibits a red shift to lower frequency in its infrared spectrum (Infrared Spectroscopy[B], James M. Thompson, 2018, P 60). As for HOF-BTB@AAO membrane, there should be some interactions between AAO disk and the -COOH group of HOF-BTB, offsetting the above-mentioned red shift in spectrum. Although the exact interaction cannot be definitively determined due to the current limitation in scientific technology, this speculation can be supported by the observed broadened band at 1606 cm^{-1} .

5. The gas separation results need to be well organized. (1) In Figure 4b, the x-axis is C₃H₆ permeability, which needs a clear membrane thickness for calculation; however, there is no such

information on membrane thickness. For supported membranes, permeance is a more common and accurate expression. (2) Figure S23, the testing pressure is below 1 bar, how is it possible? If the pressure difference is the driving force, it should be clearly marked.

Response: Thank this reviewer for the constructive suggestion.

(1) “permeability” has been revised to “permeance” in Figure 4b and other places. For the thickness of the membrane, we have the cross-sectional SEM images of these membranes in supporting information (Figure R10), from which we can tell the thickness is about 105 μm . To avoid any confusion, we now add text information in Section S3 in the revised supporting information.

(2) In Figure S23, the values of abscissa are the partial pressures of a single gas, so the total pressure is high than 1 bar (molar ratio of C_3H_6 : $\text{C}_3\text{H}_8=1:1$). We thank this reviewer for raising this question, now the label of abscissa has been changed to “partial pressure” to avoid causing any possible confusion. (Figure R11).

Figure R10. The thickness of a) AAO disk, b) HOF-BTB@AAO membrane, and c) BTB@AAO membrane.

Figure R11. Pressure dependence of $\text{C}_3\text{H}_6/\text{C}_3\text{H}_8$ separation performance for HOF-BTB@AAO membrane at 27 °C with equimolar $\text{C}_3\text{H}_6/\text{C}_3\text{H}_8$ feed flow.

6. For membrane fabrication, could the author explain why 5-cycle leads to a continuous membrane, but 10-cycle could not? If the “dissolution of the crystallized HOF-BTB” is the reason, there should be a continuous membrane with defects, but we can only find some particles like the 2-cycle sample.

Response: The small defects in HOF membrane are difficult to be detected by SEM. The HOF solution not only covered the surface of the AAO, but also filled into the AAO channel to form continuous membrane. SEM images can only provide information on the film continuity on AAO surface rather than the continuity in the channel. However, the appearance of newly emerged particles on 10-cycle AAO surface suggest the possibility of a dissolution and recrystallization process occurring as the fabrication procedure was repeated beyond five cycles. Furthermore, we employed gas permeation experiments to validate the membrane’s continuity. Based on the

experimental results, we find that 5-cycle membrane shows the best gas separation performance.

7. Cross-sectional SEM images are needed to show the thickness of the HOF membrane, where AAO support thickness should not be included.

Response: We have collected the cross-sectional SEM images to illustrate the thickness of HOF layer on the AAO surface. As shown in Figure R12, the thickness is approximately 126 nm. For better clarity, we have now also added textual description in revised manuscript.

Figure R12. The cross-sectional SEM image of HOF-BTB@AAO membrane. The thickness of HOF-BTB layer is about 126 nm.

Reviewer #3 (Remarks to the Author):

The authors presented a method to produce HOF membranes on various types of substrates. They showed the membrane has nice separation performance for C₃H₆ and C₃H₈. However, some issues need to be resolved properly before it can be considered for acceptance in Nature Communication.

1. The authors emphasized that the study is about HOF in solution state, which is quite misleading. They are HOF nanoparticles. In many cases, these nanoparticles tend to aggregate and cannot form stable colloid in solution. Rather than write something like solution state, it would be better to say forming stable colloid in solution and study the properties of the colloid.

Response: Thank the reviewer for this constructive suggestion. We agree with reviewers' idea that dissolving HOF in solution gave rise to HOF nanoparticles. Different from some conventional top-down nanoparticle fabrication, this HOF nanoparticles can even be obtained through gently shaking by hand for 20 s. Indeed, as mentioned by this reviewer, many nanoparticles tend to aggregate or grow in ambient environment. However, the obtained HOF colloid in this work exhibits remarkable long-term stability. The Cryo-electron microscopy (Cryo-EM) and Dynamic Light Scattering (DLS) measurement reveals the presence of two kinds of particles in the solution with the size of ~10 nm and ~137 nm (Figure R13a-13b). This colloid gradually transforms into monodisperse colloid with size around 100 nm after being left in ambient environment for one week, and no precipitation is observed even after one month. We think there is a balance between HOF dissolution and aggregation in solvent, which finally reach equilibrium and yield a stable colloid with size about ~100 nm (Figure R13c). We have changed the statement of "HOF solution" to "HOF colloid" in the revised manuscript.

Figure R13. a) Cryo-Transmission electron microscopy (Cryo-EM) image and b) Dynamic Light Scattering (DLS) derived particle size distribution of 5 mg/mL HOF-BTB in freshly prepared DMF solution and c) 5 mg/mL HOF-BTB in DMF solution after standing for 7 days (the experiments were repeated twice to ensure the reliability of results).

2. The authors claimed that they use 3D-ED technique to determine the crystal structure. However, I didn't find any data statistics regarding the structure determination process, such as R values, data quality, etc. Only unit cell parameters were provided. The space group is not presented at all Results Response: 3D electron diffraction technology can generate a file containing the Miller Indices and deduce unit cell information. The result shows that the unit cell determined through cryo-electron microscopy is consistent with that of the bulky HOF as described by single-crystal X-ray determination (SCXRD). However, we cannot further refine the atom positions due to the low resolution and the very large unit cell ($V = 42881.3 \text{ \AA}^3$). To provide more information regarding the structure determination process, we add Space group, N_{total} , Completeness, Resolution, and R_{int} value in Table S2 in the revised Supporting Information.

Table R1. Crystallographic unit cell of HOF-BTB obtained through 3D-ED and SCXRD

Source	3D-ED ^a	SCXRD ¹
CCDC number	-	1400566
$a/\text{\AA}$	31.026	31.419(6)
$b/\text{\AA}$	31.843	30.116(6)
$c/\text{\AA}$	44.219	45.320(9)
$\alpha/^\circ$	90	90
$\beta/^\circ$	90	90.412(2)
$\gamma/^\circ$	90	90
Space group	$I2^b$	$I2$
N_{total}	12561	96106
Completeness/%	49.0	99.8
Resolution/ \AA	1.20	0.77
$R_{\text{int}}/\%$	25.25	5.96

^a: reflection intensity extraction was conducted by the program XDS² with the aid of Coeus³.

^b: the space group was determined by the program XPrep.

3. "The larger particle size and improved colloid stability of HOF-BTB over amorphous BTB" BTB dissolved in solution is not amorphous. Amorphous is in solid state, not in solution.

Response: We thank this reviewer for pointing out this misleading description. Amorphous BTB here refers to the amorphous BTB ligand dissolved in DMF. In order to eliminate ambiguity, we change this sentence as “the larger particle size and improved stability of HOF-BTB collide over BTB ligand solution”, and all the other places with this statement were changed accordingly.

4. “there may be segments of supramolecular assemblies in HOF-BTB solution”

Do you mean some clusters of HOF-BTB or some precursor of HOF-BTB? Could you estimate how large are the clusters or segments and why they can stabilize the colloid?

Response: Yes, the larger particle size and improved stability of HOF-BTB colloid over BTB solution imply that there may exist small HOF-BTB clusters (or we can say HOF nanoparticles) in solution. This speculation was then verified by Cryo-Transmission electron microscopy (Cryo-EM) image and Dynamic Light Scattering (DLS) measurement, which shows the presence of nanosized fragments ranging from 10 nm to 200 nm (Figure 14).

The higher zeta potential of HOF-BTB colloid (33.12 mV) than BTB solution (15.65 mV) may be caused by the ordered arrangement of -COOH groups on the surface of HOF nanoparticles. The partial deprotonation of these -COOH groups caused repulsion between HOF particles and prevented them from aggregating.

Figure R14. (a) Cryo-Transmission electron microscopy (Cryo-EM) image and (b) Dynamic Light Scattering (DLS) derived particle size distribution of 5 mg/mL HOF-BTB in fresh prepared DMF solution (the experiments were repeated twice to ensure the reliability of results).

5. “This difference may be due to the interaction between H₂O molecules and the carboxyl groups of free BTB molecules in solution”

What kind of interaction does the authors refer in here? Hydrogen bonding interaction? Also where does H₂O molecules comes from? The system is in DMSO solution.

Response: The peak around 3.3 ppm indicates the existence of water in the DMSO-D₆ solvent, and the significant cross-peak at 13.1 ppm and 3.3 ppm suggests the existence of hydrogen bonding interaction between H₂O molecules and the carboxyl groups in BTB ligand solution. We have revised the statement accordingly in the main text.

6. “(HRTEM) image observes the spatial frequencies characteristic of $(0\ 7\ 7)$, $(0\ 7\ 7)$, $(0\ 7\ 7)$, $(0\ 7\ 7)$, $(0\ 0\ (14))$, and $(0\ 0\ 14)$ planes of the HOF-BTB structure”

The indexing is wrong. There is no such systematic absences in crystallography. Either the zone axis is wrong or the unit cell is incorrect. Please take an electron diffraction pattern in the same orientation.

Response: The below figures represent the experimental Fast Fourier transform (Figure R14a) image and simulated electron diffraction pattern derived from single-crystal x-ray diffraction (SCXRD) dataset in the $[1\ 0\ 0]$ direction (Figure R14b). These figures illustrate that the assigned

lattice planes are consistent with each other in both the distance and angle. The absence of some diffraction, such as (0 2 0) and (0 2 2), is probably attributed to the dynamical effect. This effect can lead to the variations in the diffraction spot intensities as the sample thickness changes, causing the intensity difference between experimental Fast Fourier transform image and simulated electron diffraction pattern⁴. So we are confident on the indexing presented in our manuscript. To eliminate any potential doubts from readers, we have now added the simulated electron diffraction pattern derived from SCXRD into the revised supporting information Figure S7.

Figure R15. a) Experimental Fast Fourier Transform (FFT) image of HOF-BTB and b) simulated diffraction pattern image calculated from HOF-BTB SCXRD data through SingleCrystal software.

7. In Figure 2 (e), xyz axis is not very useful in reciprocal space. Please use a^* , b^* and c^* axis

Response: Thank this reviewer for the careful reading. We have changed it to a^* , b^* and c^* axis.

Figure R16. Projection of a reconstructed 3D reciprocal lattice along c^* direction.

8. Figure s3 d has some problems of overlapping text.

Response: This problem has been addressed.

Figure R17. PXRD patterns of simulated HOF-BTB and HOF-BTB@Cu membrane.

9. Add a, b and c axis in the figure S17

Response: The cell axes have been added in figure.

Figure R18. Crystal structure of PFC-1. a) The structure of building block; b) the connection of adjacent building blocks; c-e) The packing structure seeing from different directions with highlighting the geometry and surface of channel.

10. Figure 2(e) clearly shows the angle between a^* and b^* is not 90 degree. However, in Figure S7, all the angles become around 90 degree. Also, the 3D reciprocal space in Figure S7 is not from a single crystal. Please determine the correct unit cell. Also provide the systematic absence to determine the space group

Response: Thank this reviewer for the careful reading. In the Figure 2e of previous manuscript, the image of reconstructed 3D reciprocal lattice was not presented along a^* , b^* , or c^* axis, so the angle between axis cannot be well distinguished. In the revised manuscript, we have included the image seeing along c^* axis to eliminate this potential source of confusion (Figure R19).

Uploading the 3D-ED data to Xprep software, the examination of the space group immediately suggested that the structure should be either I or P centered. Referring to the structure determined by SCXRD, we opted for the I lattice, leading to the suggestion of space group $I2$, $I2/m$, and Im with the CFOM values as 11.66, 7.30, and 14.51, respectively. The mean value of $|E^*E - 1|$ (0.949) falls between the value for a centrosymmetric space group (0.968) and noncentrosymmetric one (0.736). Consequently, we cannot determine whether it is centrosymmetric or not. The selection of $I2$ space group gave rise to the unit cell of $a = 31.026 \text{ \AA}$, $b = 31.843 \text{ \AA}$, $c = 44.219 \text{ \AA}$, $\alpha = \gamma = 90^\circ$, $\beta = 90^\circ$, which perfectly matches with the unit cell determined by X-ray diffraction. More details on the analysis of the systematic absence used to determine the space group can be found in the output 3D-ED.prp file, which we have included as a supporting information for review.

Figure R19. Projection of a reconstructed 3D reciprocal lattice along c^* axis from the 3D-ED dataset.

11. Please add the crystallography data statistics table.

Response: We have included Space group, N_{total} , Completeness, Resolution, and R_{int} value in reversed Supporting Information Table S2. For more details, please see our response to comment 2 of this reviewer.

12. Please attach the cif file and upload it to CCDC database and provide the corresponding CCDC number.

Response: 3D electron diffraction technology can generate a file containing the Miller Indices and deduce unit cell information. However, we cannot further refine the atom positions due to the low resolution and the very large unit cell ($V = 42881.3 \text{ \AA}^3$). Therefore, we cannot provide a cif file with atom refinement details at current stage.

CCDC number of HOF-BTB determined by single-crystal x-ray diffraction (CCDC 1400566) have been added into Table S2 in Supporting Information and the editing were highlighted as yellow.

13. Figure S3 showed the PXRD patterns of HOF on various types of substrates. All of them showed very poor crystallinity. For HOF on PET-ITO, the small peaks between 10 and 20 degree are not HOF-BTB. I suggest to remove this. PXRD for membrane on other substrate showed very few peaks and they are very wide. I am not even sure if they are HOF-BTB or not. Probably on HBG might be HOF-BTB. But on ITO and Cu, only the first peak matches and only one peak is available. So you cannot say it is not HOF-BTB but it would be quite uncertain to say it is HOF-BTB.

Response: We thank this reviewer for the constructive suggestion. We have optimized the synthetic condition, resulting in HOF-BTB membrane with improved crystallinity. The experimental PXRD patterns match better with the simulated curves. The data have been updated in the revised supporting information.

Figure R20. PXRD patterns of a) simulated HOF-BTB, ITO glass, and HOF-BTB@ITO membrane, b) simulated HOF-BTB, HBG glass, and HOF-BTB@HBG membrane, c) simulated HOF-BTB, ITO-PET substrate, and HOF-BTB@ITO-PET membrane, d) simulated HOF-BTB, Cu sheet, and HOF-BTB@Cu membrane.

14. Why HOF membrane can have good separation performance for C₃H₆/C₃H₈? Are there any explanations for the separation mechanism?

Response: Based on solution–diffusion mechanism, membrane's separation selectivity is ascribed to the synergy of solubility-selectivity and diffusion-selectivity. Solubility-selectivity mainly originates from the differences in their equilibrium adsorption, which is thermodynamically controlled permeation and dependent on the condensability of the penetrant molecules, as well as their specific affinity to the membranes. The similar adsorption behavior isotherms (Figure R21) and heat of sorption values (Figure R22, \sim 26.6 kJ/mol) of HOF-BTB for C₃H₆ and C₃H₈ suggest that C₃ selectivity does not originate from preferential thermodynamic adsorption in HOF-BTB. It means this HOF membrane's selectivity is not based on solubility-selectivity mechanism.

As for diffusion mechanism, the diffusivity of molecules through microporous framework membranes is a kinetically controlled process and depends on the size/shape differences between the penetrant molecule and micropores, properties of the inner pore surface, and gas condensability. HOF-BTB has undulated microchannels with pore limiting diameter (PLD, Figure R23) of 6.25 Å (calculated by Zeo++ program), which is larger than three-directional molecular size of C₃H₆ (6.24×5.081×4.006 Å³) and smaller than molecular size of C₃H₈ (6.505×4.365×4.018 Å³). So, we think this HOF membrane's selectivity is based on diffusion-selectivity mechanism through molecular sieving diffusion.

Figure R21. Gas sorption isotherms of C_3H_6 and C_3H_8 for HOF-BTB at 273K and 298 K.

Figure R22. The heat of sorption of C_3H_6 (red) and C_3H_8 (black) for HOF-BTB.

Figure R23. The illustration of pore limiting diameter (PLD).

15. Please obtain the size distribution of HOF-BTB nanoparticles before and after the dispersion process.

Response: The optical microscope photograph of HOF-BTB shows micron-scale particles about $100\ \mu\text{m}$, while Cryo-electron microscopy (Cryo-EM) and Dynamic Light Scattering (DLS) measurements of HOF-BTB solution show the presence of nanosized fragments ranging from 10 nm to 200 nm (Figure R24).

Figure R24. A) Optical microscope image of powdery HOF-BTB; b) Cryo-Transmission electron microscopy (Cryo-EM) image of 5 mg/mL HOF-BTB in DMF solution; and c) Dynamic Light Scattering (DLS) measurement of DMF solution with HOF-BTB (the experiments were repeated twice to ensure the reliability of results).

16. Again dissolve here is misleading. HOF-BTB are nanoparticles and able to disperse well in organic solvent.

Response: We thank this reviewer for pointing out this confusion. Combined with his/her above comment 1, the related statement has been changed as “dispersion of HOF-BTB in DMF yields a colloid solution”.

Reference

- (1) Zentner, C. A.; Lai, H. W.; Greenfield, J. T.; Wiscons, R. A.; Zeller, M.; Campana, C. F.; Talu, O.; FitzGerald, S. A.; Rowsell, J. L. (2015). High surface area and Z' in a thermally stable 8-fold polycatenated hydrogen-bonded framework. *Chem. Commun.* 51 (58), 11642-11645.
- (2) Wang, B.; Rhauderwick, T.; Inge, A. K.; Xu, H.; Yang, T.; Huang, Z.; Stock, N.; Zou, X. (2018). A Porous Cobalt Tetrakisphosphate Metal-Organic Framework: Accurate Structure and Guest Molecule Location Determined by Continuous-Rotation Electron Diffraction. *Chem. Eur. J.* 24 (66), 17429-17433.
- (3) Chen, P.; Liu, Y.; Zhang, C.; Huang, F.; Liu, L.; Sun, J. (2021). Crystalline Sponge Method by Three-Dimensional Electron Diffraction. *Frontiers in Molecular Biosciences* 8, 821927.
- (4) C. Barry Carter, D. B. W., *Transmission Electron Microscopy Diffraction, Imaging, and Spectrometry*. 2016, p 284-287.

Reviewers' Comments:

Reviewer #1:

Remarks to the Author:

The authors have made adequate modification for the manuscript and have responded all my concerns. I think it is now meet the criteria for publication on Nature Communications.

Reviewer #2:

Remarks to the Author:

The authors have addressed the issues. It can be published.

Reviewer #3:

Remarks to the Author:

The manuscript was improved a lot and most of my concerns were resolved. However, there are still some issues related to structure and HRTEM image of HOF-BTB.

1. The author showed the 3D reciprocal lattice, although the lattice is somehow distorted and not well arranged, the periodicity showed the unit cell. The author can further show the $hk0$, $0kl$ and $h0l$ by slicing through the 3D reciprocal lattice. By doing so, the space group can also be determined from the systematic absences viewed from these planes. In most cases, we do not rely on the results given by xprep to determine the space group.

2. The FFT of the HRTEM image is still problematic. As shown in the simulated image in Figure S7 (b), reflection (020) and $(0-20)$ are also strong spots (even stronger than 077 and $07-7$). If the indexing is correct, then these reflections should also appear in the FFT image, which is not presented in Figure 2 and Figure S7 (a). There are some many other spots which are stronger and they did not exist in the FFT image at all.

3. Usually to take an HRTEM image for a porous material, the easiest direction is along the direct where pore can be observed. The HRTEM image in Figure 2 is taken along a axis while the actual pore is located along c axis. The author can show some electron diffraction patterns taken along zone axis to support their claim. It is not necessary to use an HRTEM image. Imaging suffers from many influencing factors. A poor quality HRTEM image doesn't show much additional information.

4. One plot for HOF-BTB along c axis to show the pore shape and size should be added.

Response to Reviewers' comments:

The manuscript was improved a lot and most of my concerns were resolved. However, there are still some issues related to structure and HRTEM image of HOF-BTB.

1. The author showed the 3D reciprocal lattice, although the lattice is somehow distorted and not well arranged, the periodicity showed the unit cell. The author can further show the $hk0$, $0kl$ and $h0l$ by slicing through the 3D reciprocal lattice. By doing so, the space group can also be determined from the systematic absences viewed from these planes. In most cases, we do not rely on the results given by xprep to determine the space group.

Response: We thank the reviewer for constructive suggestion. The 2D slices of $0kl$, $h0l$ and $hk0$ are shown as Figure R1. The systematic absence conditions are: $0kl$: $k+l=2n+1$; $h0l$: $h+l=2n+1$; $hk0$: $h+k=2n+1$; $0k0$: $k=2n+1$, which is matched with that of $I2$ space group (two more rules $h00$: $h=2k+1$ and $00l$: $l=2k+1$ cannot be deduced based on the images viewing from these three directions).

Figure R1. The 2D slice cut of $0kl$ (a), $h0l$ (b), and $hk0$ (c) from the 3D reciprocal lattice.

2. The FFT of the HRTEM image is still problematic. As shown in the simulated image in Figure S7 (b), reflection (020) and $(0-20)$ are also strong spots (even stronger than 077 and $07-7$). If the indexing is correct, then these reflections should also appear in the FFT image, which is not presented in Figure 2 and Figure S7 (a). There are some many other spots which are stronger and they did not exist in the FFT image at all.

Response: We gratefully thank the reviewer for valuable remarks. The lack of some strong spots is an open question in our work, but we infer this phenomenon may be caused by 1) the HRTEM image is based on an ultrathin crystal layer, rather than a block single crystal which simulated images deduced from, 2) the presence of defects, dislocations, or stacking faults in the crystal lattice which caused lower crystallinity than single crystal sample, 3) the lattice plane with large d value (such as 020 and $0-20$) contains more atom and therefore strong noise, bringing blurry reflection spot. Considering the poor quality of FFT image, we use magnified TEM image featuring distinct diagonal lattice stripes (Figure R2) instead of FFT image to access the crystallinity of colloids. Meanwhile, the discussion regarding lattice planes based on the FFT of the HRTEM image was also removed in the revised manuscript. The corresponding revision on the related discussion has been implemented in the revised manuscript, with the edits highlighted in yellow.

Figure R2 Cryo-Transmission electron microscopy (Cryo-EM) images of 5 mg/mL HOF-BTB in DMF solution. (insert) enlarged image of the red-boxed region.

3. Usually to take an HRTEM image for a porous material, the easiest direction is along the direct where pore can be observed. The HRTEM image in Figure 2 is taken along a axis while the actual pore is located along c axis. The author can show some electron diffraction patterns taken along zone axis to support their claim. It is not necessary to use an HRTEM image. Imaging suffers from many influencing factors. A poor quality HRTEM image doesn't show much additional information. Response: We are grateful for this reviewer's valuable suggestion. Poor quality FFT image has been removed from manuscript. For more details, please see our response to comment 2.

4. One plot for HOF-BTB along c axis to show the pore shape and size should be added. Response: This image has been added into revised supporting information (Figure R3).

Figure R3. The packing structure of HOF-BTB along c axis.

Reviewers' Comments:

Reviewer #3:

Remarks to the Author:

The authors resolved all the concerns and questions. I have no more comments.